# Impact of Population Aging on Carbon Emissions in China: An Empirical Study Based on a Kaya Model

**DOI:** 10.3390/ijerph20031716

**Published:** 2023-01-17

**Authors:** Hua Xiang, Xueting Zeng, Hongfang Han, Xianjuan An

**Affiliations:** 1Labor Economics, School of Labor Economics, Capital University of Economics and Business, Beijing 100072, China; 2Institute of Population Economics, Capital University of Economics and Business, Beijing 100072, China

**Keywords:** China’s aging population, carbon emissions, Kaya model, consumption–production side, threshold effect

## Abstract

As the world’s largest developing country, China is facing the serious challenge of reducing carbon emissions. The objective of this study is to investigate how China’s aging population affects carbon emissions from the production and consumption sides based on an improved Kaya model. The advantage of the Kaya model is that it links economic development to carbon dioxide generated by human activities, which makes it possible to effectively analyze carbon emissions in relation to the structure of energy consumption and human activities. Based on different energy consumption structures and technological innovation levels, a threshold effect model is constructed. The results show that: (1) There is an inverted U-shaped curve relationship between population aging and carbon emissions in China. (2) Energy consumption structure and technological innovation thresholds can be derived for the impact of population aging on carbon emissions, with thresholds of 3.275 and 8.904 identified, respectively. (3) Population aging can reduce carbon emissions when the energy consumption structure does not exceed the threshold value. (4) There is no significant intervention effect of technological innovation on the relationship between population aging and carbon emissions. Based on the research results, some countermeasures and suggestions to reduce carbon emissions are proposed.

## 1. Introduction

In recent years, the need to reduce carbon emissions has become a global concern [1]. China has made low-carbon economic development a basic national policy and has committed to achieving “peak carbon” by around 2030 [2]. Currently, in the context of China’s commitment to high-quality economic development, economic growth must rely on productivity improvements, especially total factor productivity, to maintain a reasonable growth rate for China’s economy. However, China’s ageing population is expected to exceed 300 million in 2025, accounting for more than 20% of the total population, and is entering a moderate ageing phase. This will have an impact on labor supply and labor productivity, affecting China’s total factor productivity, which is currently only about 40% that of the US. Low productivity will not only severely constrain China’s ability to innovate in energy-saving and emission-reduction technologies but will also exacerbate carbon emissions in the production process. The ageing of the population has implications not only for the supply of labor, but also for the propensity of people to consume on the demand side. The reason for this is that an ageing society tends to be a low-desire society, which significantly constrain consumption and curb carbon emissions from the consumption side. Previous studies have shown that the energy consumption structure, the technology level and the population structure are closely related to the carbon emissions of a country [3]. In light of the increasing age of China’s population, this paper focuses on the impact of China’s aging population on carbon emissions from the production and consumption sides. Since the aging process of China’s population accelerated significantly from the end of the 19th century, the proportion of people aged over 65 years has reached 16.8 percent in 2021, an increase of 6.8 percentage points from 10 percent in 1999. In the interim, the rapid development of China’s economy has resulted in serious pollution of the environment [4]. In the context of increasing population aging and the necessity of changing China’s economic development model to low-carbon and technology-based development, it is of great practical significance to analyze the impact of China’s population aging on carbon emissions from the production and consumption sides. So, what is the correlation between carbon emissions and population aging in China and its provinces? This is an interesting question worthy of in-depth study. Using an improved Kaya model to empirically analyze the relationship between population aging and carbon emissions in China, focusing on its east-central and west regions, can contribute to clarifying the carbon reduction pressure in China. Moreover, use of the model can help policy makers formulate effective emission-reduction measures to facilitate the achievement of China’s mid- and long-term goals for carbon reduction and carbon peaking.

China cannot avoid pursuing a green transformation of its economy given the pressures of an aging population. The issue of economic development is ultimately a matter of consumption and production. Thus, exploring the impact of China’s aging population on carbon emissions from the production and consumption sides is of great practical value to help address the current challenges affecting China’s economic development. However, few studies have explored the impact of population aging on carbon emissions in China from the perspective of the relationship between production and consumption.

Many studies have focused on the measurement of carbon emissions and the relationship between population aging and carbon emissions. Since the 1980s, a wealth of research has been conducted on the issue of population aging and carbon emissions. The IPCC guidelines highlight four main sources of carbon emissions: energy sources (ES); industrial processes and product use (IPPU); agriculture, forestry and other land use (AFOLU); and waste emissions. Among these, more than 80% of the world’s carbon emissions come from ES and IPPU [5] and more than 5% of carbon emissions come from AFOLU [6]. The studies referred to above focus on measuring carbon emissions from the perspective of production. However, measuring carbon emissions from the production side alone does not accurately reflect the relationship between human activity and carbon emissions. Therefore, a number of researchers have conducted in-depth studies on carbon emissions focusing on consumption, with households as the research target. Such studies have included calculations of the total carbon emissions of households involving multiplying different categories of household expenditure with the carbon intensity per unit of GDP of the corresponding sector [7]. These studies provide a useful basis for this study in terms of carbon emission measurement.

With respect to research perspective and pathways, Dalton [8] explored the impact of population aging on carbon emissions in the United States from 2000 to 2100 based on a population–environment technology model. The study indicated that population aging will significantly reduce the level of carbon emissions in the United States in the future. Jiang [9] used a Kaya model to empirically test the relationship between the population structure and carbon emissions using life-cycle theory based on cross-country panel data for 26 OECD (Organization for Economic Cooperation and Development) member countries. Moreover, Wang [10] explored the relationship between population aging and carbon emissions at the theoretical level by constructing an overlapping generational model. Yang [11] quantitatively analyzed the impact of population aging on carbon emissions in China based on the STIRPAT model. Rafaj [12] quantified the impact of population aging on carbon emissions in China at the micro level based on an improved Kaya model.

Since the 1990s, research on population aging and carbon emissions has gradually increased. Existing research perspectives interpret the relationship between population aging and carbon emissions in three main alternative ways. These interpretations include that population aging exacerbates carbon emissions, that population aging suppresses carbon emissions, and that there is an inverted U-shaped curve relationship between population aging and carbon emissions. Li [13] incorporated the age structure of the population into an IPAT model and suggested that population aging suppresses carbon emissions in the long run. Hong [14] found that accelerated population aging in China suppressed carbon emissions based on an empirical analysis of the relationship between population structure and carbon emissions in China from 1995 to 2007. Shen [15] reached the same conclusion using a systematic generalized-moments estimation econometric model analysis based on interprovincial panel data for China from 1995 to 2012. In contrast, Roy [16] used a STIRPAT extended model to study data for Hunan Province from 1985 to 2007 and concluded that population aging can significantly drive carbon emissions. There are many more studies that argue that population aging contributes significantly to carbon emissions [17]. In addition, many researchers suggest that there is an inverted U-shaped curve relationship between population aging and carbon emissions [18]. Some researchers have suggested that the relationship between population aging and carbon emissions follows a U-shaped curve [19].

The above studies have explored the relationship between population aging and carbon emissions from macro and micro perspectives, respectively, but few studies have analyzed the impact of population aging on carbon emissions from a production–consumption side linkage perspective based on an improved Kaya model. There are several studies that have involved differentiated investigations of China and its eastern, central and western regions. Thus, using an improved Kaya model to analyze the impact of population aging on carbon emissions from a production–consumption side linkage perspective is necessary to accurately identify the relationship between population aging and carbon emissions in China. In relation to existing studies, the contributions of this paper include the following: (1) Based on the environmental Kuznets hypothesis, the relationship between population aging and carbon emissions was explored by region using inter-provincial panel data for China from 2000 to 2019. (2) A threshold effect model was constructed to analyze whether there is a consumption structure or technological innovation effect of the impact of population aging on carbon emissions in China.

In light of the above analysis, an improved Kaya model was used to explore the impact of population aging on carbon emissions in China from a production and consumption side perspective. A threshold effect analysis was also performed. The remainder of this paper is arranged as follows: Section 2 describes the materials and methods used; Section 3 presents the results; Section 4 provides a discussion of the findings; Section 5 presents the paper’s conclusions and suggestions for future research.

## 2. Materials and Methods

### 2.1. Overview of the Study Area

China is the world’s largest developing country in terms of economic output, with a population size of over 1.4 billion people. Mainland China contains 31 provincial-level administrative regions, which are divided into three regions: East, Central and West. Among them, the eastern region consists of 11 provinces (cities): Beijing, Tianjin, Hebei, Liaoning, Shanghai, Jiangsu, Zhejiang, Fujian, Shandong, Guangdong and Hainan; the central region includes 10 provinces (regions): Shanxi, Inner Mongolia, Jilin, Heilongjiang, Anhui, Jiangxi, Henan, Hubei, Hunan, Guangxi; and the western region includes 9 provinces (regions): Sichuan, Guizhou, Yunnan, Tibet, Shaanxi, Gansu, Qinghai, Ningxia, Xinjiang.

With the rapid development of China’s economy and society, China’s total carbon dioxide emissions reached 6.65 billion tons in 2021, an increase of 1.50 billion tons (29.12%) from 5.15 billion tons in 2012, and 4.5 billion tons (209.30%) from 2.15 billion tons in 2002 (Figure 1). However, China has now entered a new development period. The low-carbon transformation of economic development is the key to achieving the sustainable and high-quality development of China’s economy. Unfortunately, China’s demographic dividend is gradually disappearing, and the aging of the population is intensifying. Specifically, the dependency ratio of China’s elderly population reached 20.8% in 2021, an increase of 8.1 percentage points from 12.7% in 2012, and double the 10.4% of 2002 (Figure 1).

Figure 2 indicates that both carbon emissions and population aging were significantly higher in the eastern region than in the central and western regions during 2000–2019, but that the growth rate was lower than in the central and western regions. The reason for this observation is that China’s economic support policies have been tilted towards the central and western regions.

China’s eastern region had the highest carbon emissions in 2000, followed by the central region, and the western region, for which emissions were relatively low. Correspondingly, the old-age dependency ratio showed a consistent trend in east and west China (Figure 3, 2000). With China’s rapid economic development, China’s region-wide carbon emissions and elderly population dependency ratios have grown exponentially (Figure 3, 2000 transforms into Figure 3, 2019).Moreover, the elderly dependency ratios in the more economically developed regions, such as Beijing, Shanghai, Shandong, Sichuan, and Chongqing, have been significantly higher than those in the economically underdeveloped regions, such as Xinjiang and Tibet. This phenomenon is mainly a reflection of the following two factors: (1) Regions with higher levels of economic development have undergone earlier industrial and service development; the population that migrated to these regions earlier is gradually increasing in age and, thus, the degree of aging is relatively greater. (2) Economically developed areas have better infrastructure and public services, which are suitable for the elderly and, thus, attract some of them to migrate inward.

### 2.2. An Improved Kaya Model

An extended Kaya model was used in this paper. Compared with the traditional Kaya model, it has the advantages that it can more accurately and comprehensively analyze the influence and marginal effect of different factors on carbon emissions, such as the population aging structure, the energy consumption structure, the employment structure and the technological innovation level, which are considered in the regression model. This approach is more consistent with the realities of economic and social development. The Kaya model was proposed by Japanese scholar Kaya in 1990 and is usually used to represent the relationship between population size, economic development level, energy consumption and carbon emissions. The traditional model specific expression is:(1)$$C_{it}=\frac{C_{it}}{Y_{it}}\frac{Y_{it}}{POP_{it}}POP_{it}$$
where *C* represents carbon emissions, *Y* represents gross product, *POP* represents population size, the subscript i represents different cities, and *t* represents different years. Then, *C/Y* represents carbon emission intensity and *Y/POP* represents per capita GDP.

Let α = C/Y and rgdp = Y/POP, then Equation (1) can be rewritten as follows:(2)$$C_{it}=\alpha_{it}rgdp_{it}{POP}_{it}$$

Taking the logarithm of both sides of Equation (2):(3)$$lnC_{it}=ln\alpha_{it}+lnrgdp_{it}+ln{POP}_{it}$$

Equation (3) shows that, under the condition of constant carbon emission intensity (α), an increase in population size and GDP per capita will be translated into an increase in carbon emissions according to a certain distribution mechanism.

However, in the process of real economic and social development, carbon emission intensity will change with changes in technological progress (TECH), energy consumption (ECON), the consumption structure (ESTR) and other factors. The calculation formula for carbon emission intensity can be expressed as follows:(4)$$\alpha_{it}=f\left(rgdp_{it},POP_{it},ECON_{it},TECH_{it},ESTR_{it}\right)$$

Furthermore, according to the income–consumption model, the consumption structure (ESTR) usually changes with the change in employment structure (EMSTR) and income level (INCOM). Owing to the large differences in economic level, industrial structure and employment structure in different regions, this paper uses regional labor force employment structure (the ratio of the number of employees in the tertiary industry to the total number of employees) and per capita income level of residents as control variables affecting carbon emissions in the carbon emission intensity measurement model. Moreover, to minimize the impact of omitted variables on the empirical results, the classical EKC hypothesis is adopted and, as proposed in previous studies, the square term of population aging (Aging^2^) and the square term of income level (INCOME^2^) are included in the econometric analysis model. After considering the above factors, the relation of the carbon emission intensity function is as follows: (5)$$\alpha_{it}=f\left(rgdp_{it},POP_{it},ECON_{it},TECH_{it},ESTR_{it},Aging_{it},Aging_{it}{}^{2},Control_{it}\right)$$
where, Control in Equation (5) includes income level (INCOM) and its square term, employment structure (EMSTR), gross product per capita (rgdp) and population size (POP). Taking the double logarithm of both sides of Equation (5), the random error term and a province dummy variable are added to estimate the influence parameter values of different regions. The specific carbon emission intensity estimation equation is as follows:(6)$$\begin{array}{l} ln\alpha_{it}=\alpha +\beta_{1}lnrgdp_{it}+\beta_{2}lnPOP_{it}+\beta_{3}lnECON_{it}+\beta_{4}lnTECH_{it}\\ \;+\beta_{5}lnESTR_{it}+\beta_{6}lnAging_{it}+\beta_{7}lnAging_{it}{}^{2}+\beta_{8}lnControl_{it}\\ \;+\theta_{i}+\varepsilon_{it} \end{array}$$

Equations (3) and (6) were further combined and simplified. The extended KAYA equation can be obtained as follows:(7)$$\begin{array}{l} lnC_{it}=\alpha +\gamma_{1}lnrgdp_{it}+\gamma_{2}lnPOP_{it}+\beta_{3}lnECON_{it}+\beta_{4}lnTECH_{it}+\\ \beta_{5}lnESTR_{it}+\beta_{6}lnAging_{it}+\beta_{7}lnAging_{it}{}^{2}+\beta_{8}lnControl_{it}+\theta_{i}+\varepsilon_{it} \end{array}$$

### 2.3. Variables and Data

#### 2.3.1. Variables

The explained variable is carbon emissions. The carbon emission accounting method jointly adopted by the Intergovernmental Panel on Climate Change (IPCC), the Office of the China Climate Change Policy Coordination Group and the Energy Research Institute of China Development and Reform Commission is used. The total amount of carbon dioxide emitted from coal, coke, gasoline, kerosene, diesel oil, fuel oil, natural gas and cement production is defined as carbon dioxide emissions in this paper [20]. The calculation formula is:(8)$$C=\overset{7}{\underset{i=1}{\sum }}EC_{i}+CS=\overset{7}{\underset{i=1}{\sum }}E_{i}\times CF_{i}\times CC_{i}\times COF_{i}\times 3.67+Q\times EF_{concrete}$$

In Equation (8), the subscript i represents seven kinds of energy, E represents the energy consumption, CF represents the heating value of energy, CC represents the carbon content, COF represents the oxidation factor, Q represents the cement production, and EF represents the carbon emission coefficient of cement production.

The core explanatory variable is population aging. The World Health Organization WTO) defines people over the age of 65 as elderly. Almost all existing studies use “the ratio of the population over 65 years old to the total population” or “the dependency ratio of the elderly” to measure the level of population aging, and this is used in this study. The dependency ratio of the elderly population over 65 years old was used to characterize the aging level (Aging) and the quadratic term of population aging (Aging^2^) was also included in the regression model.

The explanatory variables for mechanism analysis include the technological innovation level (TECH) and the energy consumption structure (ESTR). The TECH index was calculated collectively. Firstly, the eight three-level indicators (Table 1) were treated as dimensionless. Secondly, the entropy weight method was used to calculate the weight of each of the three levels. Finally, the weighted average of each index was calculated. The energy consumption structure (ESTR) was characterized by the ratio of the energy consumption of daily necessities to the energy consumption of healthcare products. The total energy consumption (ECON) was determined by the total energy consumption of each province and city (per ten thousand tons of standard coal).

The control variables were income level (INCOM) and its square term, employment structure (EMSTR), gross product per capita (rgdp) and population size (POP). Among them, income level (INCOM) was measured by the per capita disposable income level of provinces and cities. The square term of income level (INCOM^2^)was expressed by income level * income level. Population size (POP) was characterized by the size of the resident population in each province and city. The employment structure (EMSTR) was represented by the ratio of the number of employees in the tertiary industry to the total number of employees. Gross domestic product per capita (rgdp) was measured using GDP per capita adjusted for constant prices in the 1999 base period, which effectively eliminates the impact of inflation on the empirical analysis results.

#### 2.3.2. Data

The data used in this study were the provincial-level panel data for 31 provinces and cities in mainland China from 2000 to 2019, which were derived from the statistical data for energy, population, employed persons, wages and science and technology indicators published in the Annual Database by Province of the National Bureau of Statistics. The subsequent empirical analysis of the threshold effect model in this study required strictly balanced panel data. As some of the missing value types were non-deterministic values, the missing values were filled in by calculating similar mean values based on the statistical principle of plurality and the K-nearest neighbor filling method. The specific steps involved are: First, classify the known values; second, calculate the difference between the maximum and minimum values and divide by three; third, divide the values into three major categories based on the size of the difference between the values; fourth, group the missing values into the category with the largest number of samples based on the principle of plurality; and, fifth, calculate the mean value of the data belonging to the category, which is the missing value. The K-nearest neighbor filling (KNN) method was used to make up the missing values of the research samples of Tibet and Xinjiang in the process of data collation.

## 3. Results

### 3.1. Descriptive Statistical Analysis

Table 2 shows that there were 620 study samples in total, which meets the sample size requirements for panel data analysis. Considering the basic characteristics of the data, the mean value expressed logarithmically of carbon dioxide emissions was 7.451; the minimum and maximum values were 4.104 and 9.105, respectively. The mean value of per capita GDP expressed logarithmically was 7.607; the minimum value and maximum value were 3.671 and 10.351, respectively. The mean value of the logarithm of population aging was 2.512; the minimum value and maximum value were 1.435 and 3.170, respectively, and the standard deviation was 0.248, which indicates that population aging in different provinces and cities across the country showed little difference. The mean value of the technological innovation level was 7.600; the standard deviation was 2.143 and the minimum value and maximum value were 0.000 and 12.015, respectively, representing significant differences. Based on analysis of the basic characteristics of the above data, the following simple conclusion can be drawn that: Population aging has a small direct impact on carbon emissions but a substantial indirect impact on carbon emissions through technological innovation on the production side. To explore further the degree and nature of the impact of population aging on carbon emissions through the production side and the consumption side, the empirical tests and analysis performed are described below.

To ensure the reliability of the results of the subsequent empirical analysis, stata17.0 software was used to test the correlations between carbon emissions and population aging and other variables. The test results show that carbon emissions were significantly correlated with the variables of population aging, per capita economic development level, population size, total energy consumption, and energy consumption structure with a significance level of 1% (Table 3).

### 3.2. Baseline Model Estimation

The benchmark regression model mainly tests the impact of population aging, consumption structure and technological innovation on carbon emissions. To find the optimal fitting analysis model, the data was first analyzed using a random effect model, a fixed effect model and a mixed effect model, followed by use of the Hausman test. The test results showed that the fixed effect model solution provided the best-fitting solution.

#### 3.2.1. Impact of Population Ageing on Carbon Emission

Based on the basic model (7), a regression Equation (9) of population aging and carbon emissions was constructed for empirical analysis.
(9)$$lnC_{it}=\alpha +\alpha_{1}lnrgdp_{it}+\alpha_{2}lnPOP_{it}+\alpha_{3}{lnAging}_{it}\;\;+\alpha_{4}{\left(lnAging_{it}\right)}^{2}+\alpha_{5}lnControl_{it}+\theta_{i}+\varepsilon_{it}$$

All the variables in the regression model passed the multicollinearity test; there was no collinearity problem between the independent variables and the control variables. Based on the test results using the Hausman test, a fixed effect model was used for regression analysis. The regression results (Table 4) showed that the relationship between population aging and carbon emissions did not conform to the environmental Kuznets curve (EKC) hypothesis, but it clearly indicated that different levels of population aging affected the trend in carbon emissions. According to the coefficients of the regression equation, population aging in the whole country, central China and western China significantly affected carbon emissions. When the aging degree of the population reaches the first inflection point of the regression curve, carbon emissions will rise along with the deepening of the population aging degree and cause deterioration of the climate environment (Hypotheses 1 and 2 are supported). The results of the national sample regression analysis show that, if the degree of population aging increases by 1%, carbon emissions will decrease by 1.387%; The effect of income level on carbon emissions conforms to the EKC hypothesis. Upgrading of the industrial structure (lnEMSTR) can effectively restrain carbon emission. Rising population size and per capita economic growth both accelerate carbon emissions. The above empirical test results show that, with the irreversible trend in population aging in China, technological innovation and industrial structural transformation and upgrading are the only way to achieve economic development.

#### 3.2.2. Effect of Energy Consumption Structure on Carbon Emission

Based on the basic model (7), the regression Equation (10) of consumption structure and carbon emissions was constructed for empirical analysis.
(10)$$\begin{array}{l} lnC_{it}=\sigma +\sigma_{1}lnrgdp_{it}+\sigma_{2}lnPOP_{it}+\sigma_{3}lnECon_{it}+\rho_{4}lnEStr_{it}+\rho_{5}{lnControl}_{it}\\ \;+\theta_{i}+\varepsilon_{it} \end{array}$$

Table 5 shows the analysis results for the fixed-effect model. In terms of the direction of influence, under the current level of economic and technological development, there is a positive correlation between the energy consumption structure (lnEMSTR) and carbon emissions in the eastern and central regions, but there is a reverse relationship in the western region. In terms of the direction of influence, under the current level of economic and technological development, there is a positive correlation between energy consumption structure (lnEMSTR) and carbon emissions in the eastern and central regions, but there is a reverse relationship in the western region. These results are consistent with the economic and social reality. The reason for this phenomenon is that the economic development level, consumption tendency and consumption ability of the eastern and central regions are all ahead of those of the western regions. With regard to the significance level of the regression parameters, the effect between energy consumption structure and carbon emissions was not significant because carbon emissions are more affected by the consumption behavior of economic subjects [21]. The above research results show that economic development is bound to be accompanied by an increase in carbon emissions, but changing residents’ consumption concepts, advocating green consumption and improving residents’ green consumption levels can reduce the growth rate of carbon emissions, which can promote China’s economy to achieve the goal of green and sustainable development.

#### 3.2.3. Impact of Technological Innovation on Carbon Emission

Based on the basic model (7), the regression Equation (11) of consumption structure and carbon emissions was constructed for empirical analysis.
(11)$$lnC_{it}=\delta +\delta_{1}lnrgdp_{it}+\delta_{2}lnPOP_{it}+\delta_{3}lnTech_{it}+\delta_{4}lnControl_{it}+\theta_{i}+\varepsilon_{it}$$

The results of the fixed-effect model analysis (Table 6) show that technological innovation has a strong inhibitory effect on carbon emissions. According to the regression parameter values of the national sample, in the context of the same economic development speed, carbon emissions will decrease by 0.064% for every 1% increase in the level of technological innovation (lnTECH); the significance level of the result is 1%. This result indicates that technological innovation may consume a relatively large quantity of capital in the short term, while technological innovation plays an important role in accelerating the low-carbon transformation in terms of enterprises’ production mode and promoting the green development of the economy in the long run. The regression coefficient values of the control variables also show that upgrading of the employment structure will significantly reduce carbon emissions. Both population expansion and economic growth can increase carbon emissions; the relationship between income level and carbon emissions conforms to the EKC hypothesis. The above analysis results show that the optimization and upgrading of industrial structure exerts a demonstrable effect on the green development of China’s economy.

### 3.3. Threshold Effect Estimation

To further explore whether the impact of population aging on carbon emissions shows a threshold effect in terms of the energy consumption structure and technological innovation, a threshold effect test analysis is described below. 

#### 3.3.1. Threshold Value Test

Consumption structure and technological innovation were set as threshold variables, and the threshold dependence variable of population aging level was used to establish the threshold effect model. Referring to the analysis method of Yang K.J. and Yang T.T. (2017), a bootstrap test method was used to determine the number of threshold values. Comparing the significance level P of each threshold value and the value of statistic F in the test results, it was determined that there was only one threshold in the above threshold regression model. The specific results were as follows:(12)$$\begin{array}{l} lnC_{it}=\varphi_{1}lnrgdp_{it}+\varphi_{2}lnPOP_{it}\\ \;+\varphi_{3}lnAging_{it}I({\mathrm{lnESTR}}_{\mathrm{it}}\\ \;\leq k)+\varphi_{4}lnAging_{it}I\left({\mathrm{lnESTR}}_{\mathrm{it}}>k\right)+\varphi_{5}lnControl_{it}+\theta_{i}+\varepsilon_{it} \end{array}$$

Using Model 12 to test the threshold value of the consumption structure, the national analysis results show that the consumption structure has a single threshold effect on carbon emissions. The threshold value is 3.275, which passes the test for a single threshold at a level of 1% (Table 7). When the consumption structure reaches 3.275, the impact of population aging on carbon emissions will change, which indicates that the aging of the population has a different degree of impact on carbon emissions on both sides of the consumption structure around the threshold.
(13)$$lnC_{it}=\omega_{1}lnrgdp_{it}+\omega_{2}lnPOP_{it}+\omega_{3}lnAging_{it}I({\mathrm{lnTECH}}_{\mathrm{it}}\leq \;k)+\omega_{4}lnAging_{it}I({\mathrm{lnTECH}}_{\mathrm{it}}>k)+\omega_{5}lnControl_{it}+\theta_{i}+\varepsilon_{it}$$

Using model 13 to test the threshold value of technological innovation, it was found that the national data regression results showed that there was a single threshold effect of technological innovation on carbon emissions. The threshold value is 8.904, which passes the single threshold test at a 10% confidence level (Table 8). That is, when the value of the comprehensive technological innovation index reaches 8.904, the impact coefficient of population aging on carbon emissions will change, which indicates that the aging of the population has a different degree of impact on carbon emissions on both sides of technological innovation around the threshold.

In Equations (12) and (13), I (·) is an indicative function, k is a specific threshold value, $\varphi_{1},\;\varphi_{2}\;$and$\;\omega_{1},\omega_{2}$ are the parameter estimates of the impact of population aging on carbon emissions when the threshold variables meet the following conditions, which contains $lnESTR_{it}$ ≤ K, $lnESTR_{it}\;$> K, $lnTECH_{it}$ ≤ K, $lnTECH_{it}\;$> K, $\varepsilon_{it}$ is a random error term.

#### 3.3.2. Regression Results of Threshold Model

The impact of population aging on carbon emissions in different consumption structures. The regression results (Table 9) show that, in the case of exogenous technology, when the energy consumption structure (lnEMSTR) is lower than the threshold value of 3.275, the elasticity coefficient of the impact of population aging on carbon emissions is −0.154, and the confidence level is 1%. When the energy consumption structure crosses the threshold value, the elasticity coefficient of the impact of population aging on carbon emissions becomes −0.035, but the impact relationship is not significant. This indicates that the impact of population aging on regional carbon emission level is affected by the energy consumption structure, but with a lower sensitivity (the elasticity coefficients of −0.0154 and −0.035 are less than 1). Furthermore, the regression parameter value of the control variables −0.154 indicates that industrial upgrading will inhibit increase in carbon emissions, which is consistent with the influence direction of the benchmark regression model.

The impact of population aging on carbon emissions in the context of different technological innovation levels. The regression results (Table 10) show that, in terms of technological innovation at the production side, there is no obvious threshold effect of population aging on carbon emissions. However, on the other side of the threshold value, the elasticity coefficient becomes 0.052, which indicates that the regional carbon emission level is less affected by the level of technological innovation and has a lower sensitivity (the elasticity coefficients −0.004 and 0.052 are less than 1). The above results reflect the difficulty of implementing technological innovation. Only when the technology innovation level reaches a certain degree will the intensification of population aging reduce carbon emissions. In other words, to reduce the negative impact of high labor costs caused by population aging, enterprises will increase investment in technological innovation in the long run to optimize industrial structure, improve the level of the low-carbon economy and reduce carbon emissions.

## 4. Discussion

To further verify the reliability of the results of the previous empirical study, a robustness test analysis of the study is described below.

### 4.1. Sub-Sample Regression

#### 4.1.1. Impact of Aging on Carbon Emissions under the Energy Consumption Structure

Empirical analysis of the effect of population aging on carbon emissions shows that there is a threshold effect of energy consumption structure on the impact of population aging on carbon emissions in the eastern, central and western regions. The regression analysis results for the eastern region show that the elasticity coefficients of the impact of population aging on carbon emissions were 0.121 and 0.162, respectively, at the first or second stage of energy consumption structure, and the significance levels were 10% and 5%, respectively. This suggests that the current energy consumption structure in the eastern region is not suitable for an aging population society and needs to make adjustments to adapt to the development of the aging society to achieve the goal of reducing carbon emissions. The regression results for the western region show that the elasticity coefficients for the impact of population aging on carbon emissions are −0.471 and −0.312, respectively, at the first or second stage of the energy consumption structure. Moreover, the confidence level is 1% and the direction of impact is consistent with the regression results of the national data. In the central region, population aging has different effects on carbon emissions on the left and right sides of the threshold value. When the energy consumption structure is greater than the threshold value, the impact of population aging on carbon emissions is higher (Table 11). The above results indicate that population aging has different effects on carbon emissions in different energy consumption structures. The energy consumption structure and population aging need to be further coordinated and optimized.

#### 4.1.2. Impact of Aging on Carbon Emissions under Technological Innovation

The empirical analysis of the impact of population aging on carbon emissions in different levels of technological innovation in the national, eastern, central and western regions shows that (Table 12) there are no obvious technological innovation threshold effects of population aging on carbon emissions in the eastern and central regions. In the eastern region, the impact direction of population aging on carbon emissions is negative at the current level of technological innovation, which indicates that technological innovation in the eastern region is good and the degree of adaptation to population aging is high. However, the impact of population aging on carbon emissions is positive in the central region, which indicates that the relationship between population aging and the technological innovation level should be further considered, while developing the economy in the central region. There is an obvious threshold effect of technological innovation of population aging on carbon emissions in western China. However, whether the value of technological innovation is on the left or the right side of the threshold value, the impacts of population aging on carbon emissions are negative. The elasticity coefficients are −0.215 and −0.157, with confidence levels of 5% and 10%, respectively. The above analyses indicates that there is a high degree of synergy between the technological innovation level and population aging in the western region and that the current technological innovation level in this region can effectively promote green economic development in the context of population aging.

### 4.2. Independent Variable Replacement

#### 4.2.1. Impact of Aging on Carbon Emissions under the Energy Consumption Structure

In this section, replacement of the independent variable and conduct of a control test on the time effect is described. More Specifically, the squared term of population aging is used as a proxy variable for population aging due to the fact that the squared term of population aging can also capture the degree of population aging to some extent. The regression results of the model National-(1) show that there is an obvious threshold effect of the energy consumption structure on the impact of the population aging square term on carbon emissions (Table 13). The regression results of model National-(2) show that the direction and degree of effects are essentially equivalent to those of model National-(1). In addition, the regression results for the controlling time effect show that the influence direction and effect degree remain basically unchanged. The regression analysis is demonstrated to show good robustness.

#### 4.2.2. Impact of Aging on Carbon Emissions under Technological Innovation

In this section, the independent variables are replaced and the time effect is controlled. The analysis results (Table 14) show that the impact of the square term of population aging on carbon emissions has an obvious technological innovation threshold effect in the second stage, with an elasticity coefficient of −0.018, with a confidence level of 10%. The regression results controlling the time effect show that there is no obvious technological innovation threshold effect on the influence of the population aging square term on carbon emissions; the direction and degree of influence are basically consistent with the previous regression results, which indicates that the model has good robustness.

## 5. Conclusions

This study was based on Chinese provincial balanced panel data from 2000 to 2019 and used extended Kaya and threshold effect models to comprehensively analyze the impact of population aging on carbon emissions through consumption and production channels. The results showed that: (1) Population aging can effectively reduce carbon emissions. On the consumption side: (2) There is neither a significant relationship between the energy consumption structure and carbon emissions nor a threshold effect of the energy consumption structure between population aging and carbon emissions; (3) Population aging reduces carbon emissions mainly through a reduction in consumption level. On the production side: (4) Strengthening technological innovation can effectively reduce carbon emissions; (5) There is a threshold effect of technological innovation level between population aging and carbon emissions, but the effect is not significant. Consequently, the negative effect of population aging on technological innovation is not obvious. Even in the context of an irreversible trend in population aging, a reasonable economic development model and guiding policy can promote steady improvement in the national technological innovation level.

## 6. Suggestions

Based on the research undertaken, the following policy implications are highlighted: (1) Presently, in the realistic context of irreversible population aging, advocating green consumption and low-carbon consumption are the primary methods to effectively control carbon emissions on the consumption side. Citizens should increase daily physical exercise, especially middle-aged and elderly people. Only by improving the physical fitness of the whole population can the consumption of medical and health-care products be effectively reduced to fundamentally optimize the energy consumption structure and continuously reduce carbon emissions. (2) Scientific and technological innovation is a key factor in energy conservation and emission reduction. Especially in the west, central and other regions where the economic development level is less advanced, green science and technology projects relating to production should be increased. The system for training scientific and technological personnel should be improved and the policy system for technological innovation in research institutions should be strengthened, which plays an important role in promoting the achievement of the national goal of “carbon peaking and carbon neutrality”.

## Figures and Tables

**Figure 1 ijerph-20-01716-f001:**
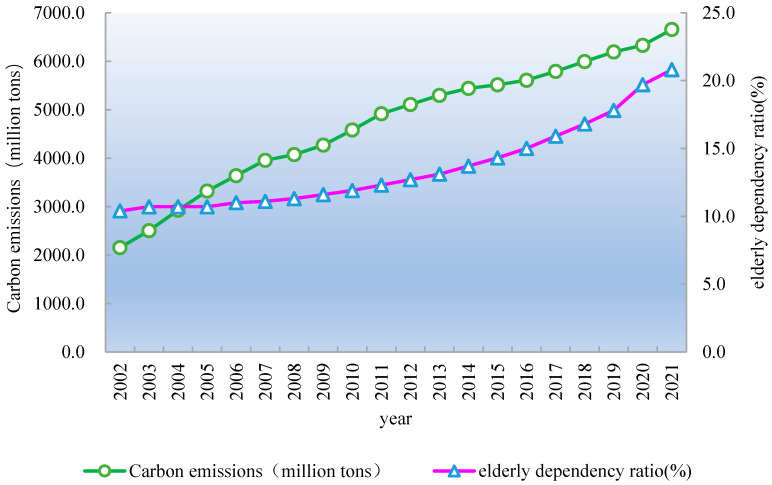
Carbon emissions and elderly population dependency ratio in China, 2002–2021.

**Figure 2 ijerph-20-01716-f002:**
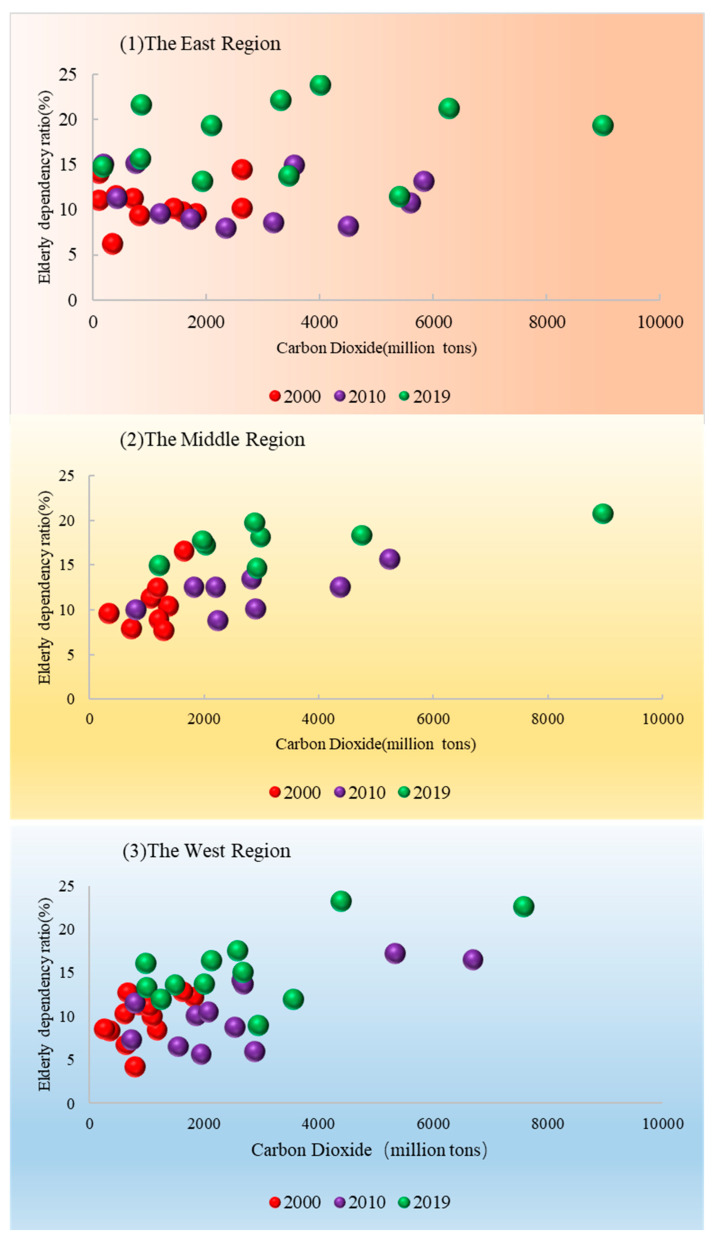
Ageing dependency ratio and carbon emissions in East Midwest Region, 2000/2010/2019.

**Figure 3 ijerph-20-01716-f003:**
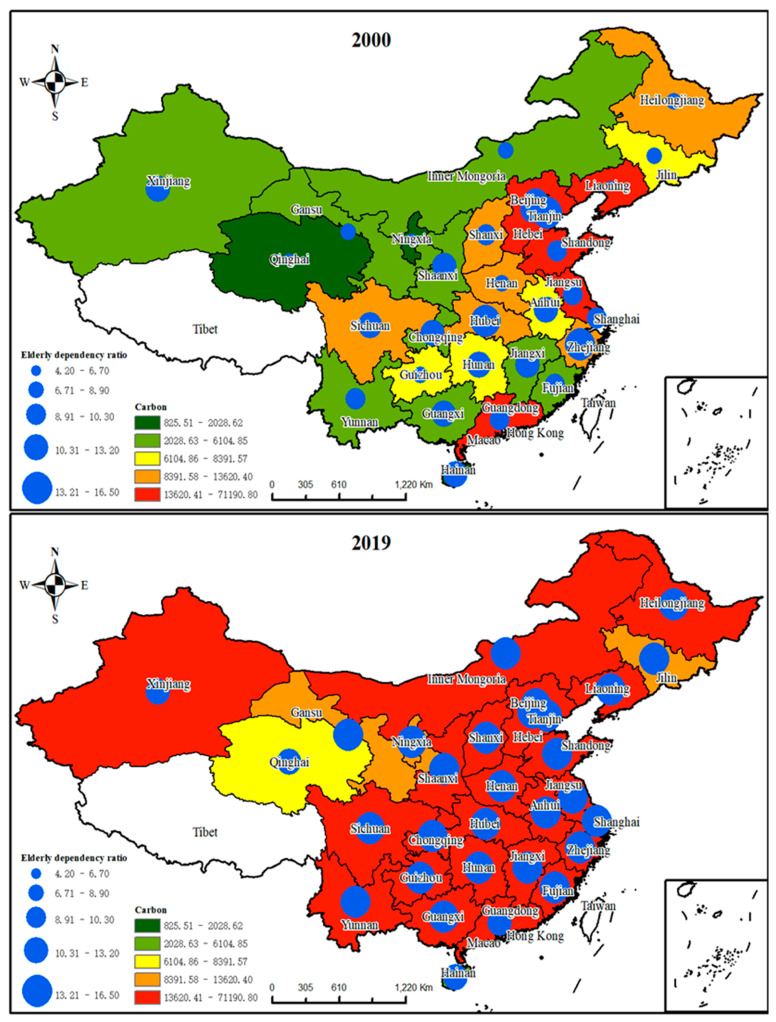
Carbon emissions and elderly dependency ratio by provincial administrative region in China in 2000 and 2019.

**Table 1 ijerph-20-01716-t001:** Comprehensive evaluation index of technological innovation.

First Level Index	Secondary Indicators	Measurements	Unit	Weights
Science And technology innovation	Innovation Inputs	Full-time equivalent of R&D personnel in enterprises above designated size	person	0.0892
		Number of R&D projects of enterprises above designated size	item	0.1381
		Funds for innovative products of enterprises above designated size	100 million yuan	0.0851
	Innovation Outputs	Three kinds of patent licensing quantity	item	0.1068
		Number of invention patents authorized	item	0.1896
		Sales revenue of innovative products of enterprises above designated size	100 million yuan	0.1237
		Technical market turnover	100 million yuan	0.1689
		Number of new product projects of enterprises above designated size	item	0.0986

Note: R&D is the abbreviation for research and development in China.

**Table 2 ijerph-20-01716-t002:** Descriptive statistical analysis.

Variable	Obs	Mean	Std. Dev.	Min	Max
lnCO_2_	620	7.451	0.882	4.104	9.105
lnAging	620	2.512	0.248	1.435	3.170
ln^2^Aging	620	6.372	1.238	2.059	10.047
lnrgdp	620	7.607	1.247	3.671	10.351
lnPOP	620	10.543	1.202	7.844	14.335
lnTECH	620	7.600	2.143	0.000	12.015
lnESTR	620	3.296	0.601	0.000	3.669
lnECON	620	8.752	1.882	0.000	10.631
lnEMSTR	620	3.587	0.295	0.000	4.388
lnINCOM	620	10.072	0.841	7.923	12.011
ln^2^INCOM	620	102.16	16.825	62.768	144.265

Note: “lnx” means the logarithm of the variable “x” and “ln^2^x” means the squared term of the variable “lnx”.

**Table 3 ijerph-20-01716-t003:** Correlation analysis of variables.

Variables	(1)	(2)	(3)	(4)	(5)	(6)	(7)	(8)	(9)	(10)	(11)
(1) lnCO_2_	1.000										
(2) lnAging	0.306	1.000									
	(0.000)										
(3) ln^2^Aging	0.317	0.997	1.000								
	(0.000)	(0.000)									
(4) lnrgdp	0.163	0.348	0.351	1.000							
	(0.000)	(0.000)	(0.000)								
(5) lnPOP	0.282	0.279	0.270	−0.482	1.000						
	(0.000)	(0.000)	(0.000)	(0.000)							
(6) lnTECH	0.399	0.591	0.589	0.590	0.366	1.000					
	(0.000)	(0.000)	(0.000)	(0.000)	(0.000)						
(7) lnESTR	0.025	0.374	0.356	0.095	0.384	0.428	1.000				
	(0.539)	(0.000)	(0.000)	(0.018)	(0.000)	(0.000)					
(8) lnECON	0.152	0.429	0.414	0.229	0.464	0.631	0.832	1.000			
	(0.000)	(0.000)	(0.000)	(0.000)	(0.000)	(0.000)	(0.000)				
(9) lnEMSTR	-0.040	0.336	0.343	0.511	-0.091	0.506	0.217	0.211	1.000		
	(0.320)	(0.000)	(0.000)	(0.000)	(0.023)	(0.000)	(0.000)	(0.000)			
(10) lnINCOM	0.263	0.460	0.468	0.714	0.011	0.792	0.210	0.318	0.736	1.000	
	(0.000)	(0.000)	(0.000)	(0.000)	(0.785)	(0.000)	(0.000)	(0.000)	(0.000)		
(11) ln^2^INCOM	0.256	0.461	0.470	0.715	0.010	0.791	0.210	0.315	0.741	0.999	1.000
	(0.000)	(0.000)	(0.000)	(0.000)	(0.808)	(0.000)	(0.000)	(0.000)	(0.000)	(0.000)	

Note: The content in parentheses indicates the significance level; “ln_x” means the logarithm of the variable “x”, “ln2x” means the squared term of the variable “lnx”.

**Table 4 ijerph-20-01716-t004:** The impact of population aging on carbon emissions.

Variables	National	The East	The Middle	The West
lnAging	−1.387 ***	−0.657	−3.344 ***	−1.303 **
	(−3.259)	(−0.752)	(−3.073)	(−2.258)
ln^2^Aging	0.282 ***	0.154	0.696 ***	0.256 **
	(3.257)	(0.902)	(3.208)	(2.008)
lnPOP	2.192 ***	2.386 ***	−1.025 **	1.641 ***
	(13.103)	(11.453)	(−2.107)	(4.782)
lnrgdp	2.184 ***	2.381 ***	−1.033 **	1.635 ***
	(13.023)	(11.401)	(−2.121)	(4.763)
lnEMSTR	−0.175 ***	−0.429 **	−0.496 ***	−0.095 **
	(−3.935)	(−2.137)	(−3.685)	(−2.053)
lnINCOM	1.260 ***	2.834 ***	6.289 ***	2.630 ***
	(6.480)	(7.330)	(9.345)	(6.708)
ln^2^INCOM	−0.154 ***	−0.233 ***	−0.240 ***	−0.199 ***
	(−12.218)	(−10.350)	(−9.236)	(−9.577)
_cons	−26.909 ***	−38.981 ***	−6.339	−25.686 ***
	(−12.645)	(−11.470)	(−1.311)	(−7.258)
N	620.000	220.000	160.000	240.000
R^2^	0.837	0.868	0.917	0.849

Note: ** indicates 5%, *** indicates 1%; _cons represents the constant term of the regression equation; the numeric in brackets indicates the standard deviation of the regression analysis; and “ln_x” means the logarithm of the variable “x”, “ln2x” means the squared term of the variable “lnx”.

**Table 5 ijerph-20-01716-t005:** The impact of consumption structure on carbon emissions.

Variables	National	The East	The Middle	The West
lnESTR	0.030	0.149	0.141	−0.029
	(0.792)	(1.370)	(1.475)	(−0.607)
lnPOP	2.037 ***	2.206 ***	−1.185 **	1.461 ***
	(13.342)	(11.443)	(−2.357)	(4.799)
lnrgdp	2.030 ***	2.209 ***	−1.185 **	1.450 ***
	(13.252)	(11.424)	(−2.355)	(4.760)
lnECON	0.014	0.012	0.008	0.022
	(1.540)	(0.698)	(0.606)	(1.616)
lnEMSTR	−0.170 ***	−0.331 *	−0.435 ***	−0.090 *
	(−3.811)	(−1.707)	(−3.415)	(−1.941)
lnINCOM	1.114 ***	2.669 ***	5.406 ***	2.571 ***
	(5.783)	(6.745)	(8.490)	(6.502)
ln^2^INCOM	−0.139 ***	−0.217 ***	−0.186 ***	−0.187 ***
	(−11.814)	(−9.829)	(−7.648)	(−9.754)
_cons	−26.126 ***	−37.329 ***	−4.700	−24.839 ***
	(−12.834)	(−11.296)	(−0.925)	(−7.549)
N	620.000	220.000	160.000	240.000
R^2^	0.836	0.867	0.911	0.846

Note: * indicates 10% significance level, ** indicates 5%, *** indicates 1%; _cons represents the constant term of the regression equation; the numeric in brackets indicates the standard deviation of the regression analysis; “ln_x” means the logarithm of the variable “x”, “ln^2^x” means the squared term of the variable “lnx”.

**Table 6 ijerph-20-01716-t006:** The impact of technological innovation on carbon emissions.

Variables	National	The East	The Middle	The West
lnTECH	−0.064 ***	−0.044 *	−0.019	−0.044 **
	(−5.299)	(−1.744)	(−1.288)	(−2.048)
lnPOP	1.940 ***	2.142 ***	−1.105 **	1.447 ***
	(12.919)	(10.880)	(−2.185)	(4.803)
lnrgdp	1.927 ***	2.133 ***	−1.113 **	1.436 ***
	(12.785)	(10.767)	(−2.201)	(4.761)
lnECON	0.018 **	0.012	0.010	0.016 *
	(2.474)	(0.690)	(0.692)	(1.804)
lnEMSTR	−0.152 ***	−0.262	−0.411 ***	−0.086 *
	(−3.475)	(−1.337)	(−3.257)	(−1.869)
lnINCOM	1.035 ***	2.608 ***	5.244 ***	2.425 ***
	(5.492)	(6.569)	(7.886)	(6.096)
ln2INCOM	−0.136 ***	−0.215 ***	−0.184 ***	−0.183 ***
	(−11.863)	(−9.716)	(−7.350)	(−9.550)
_cons	−24.306 ***	−35.798 ***	−4.567	−23.932 ***
	(−12.119)	(−10.564)	(−0.895)	(−7.267)
N	620.000	220.000	160.000	240.000
R^2^	0.843	0.868	0.911	0.849

Note: * indicates 10% significance level, ** indicates 5%, *** indicates 1%; “_cons” represents the constant term of the regression equation; the numeric in brackets indicates the standard deviation of the regression analysis; “ln_x” means the logarithm of the variable “x”, “ln^2^x” means the squared term of the variable “lnx”.

**Table 7 ijerph-20-01716-t007:** Energy consumption structure threshold.

**Single Threshold**	Threshold Value		Lower Limit	Upper Limit
	3.275		3.262	3.325
**Bootstrap Test**	RSS	MSE	F	P
	14.987	0.025	223.54	0.000

**Table 8 ijerph-20-01716-t008:** Technical innovation level threshold.

**Single Threshold**	Threshold Value		Lower limit	Upper limit
	8.904		8.824	8.912
**Bootstrap Test**	RSS	MSE	F	P
	15.221	0.025	34.72	0.080

**Table 9 ijerph-20-01716-t009:** Energy consumption structure threshold effect of population aging on carbon emissions.

Variables	National
lnAging (lnESTR ≤K)	−0.154 ***
	(−3.268)
lnAging (lnESTR ≥ K)	−0.035
	(−0.764)
lnPOP	1.537 ***
	(10.124)
lnrgdp	1.548 ***
	(10.205)
lnEMSTR	−0.079 *
	(−1.905)
lnINCOM	0.625 ***
	(3.468)
ln^2^INCOM	−0.095 ***
	(−8.329)
_cons	−16.652 ***
	(−7.972)
N	620.000
R^2^	0.987
F	531.960

Note: * indicates 10% significance level, *** indicates 1%; _cons represents the constant term of the regression equation; the numeric in brackets indicates the standard deviation of the regression analysis. “National” indicates that the empirical analysis sample is the total sample at the national level; “lnx” means the logarithm of the variable “x”, “ln^2^x” means the squared term of the variable “lnx”.

**Table 10 ijerph-20-01716-t010:** Technological innovation threshold effect of population aging on carbon emissions.

Variables	National
lnAging (lnTECH ≤K)	−0.004
	(−0.073)
lnAging (lnTECH ≥ K)	0.052
	(1.036)
lnPOP	2.080 ***
	(13.230)
lnrgdp	2.070 ***
	(13.135)
lnEMSTR	−0.150 ***
	(−3.437)
lnINCOM	1.647 ***
	(7.977)
ln^2^INCOM	−0.170 ***
	(−13.363)
_cons	−28.956 ***
	(−13.584)
N	620.000
R^2^	0.991
F	445.895

Note: *** indicates 1%; _cons represents the constant term of the regression equation; the numeric in brackets indicates the standard deviation of the regression analysis. “National” indicates that the empirical analysis sample is the total sample at the national level; “lnx” means the logarithm of the variable “x”, “ln^2^x” means the squared term of the variable “lnx”.

**Table 11 ijerph-20-01716-t011:** The threshold effect of energy consumption structure on the impact of aging on carbon emissions in eastern, central and western regions.

Variables	The East	The Middle	The West
lnAging (lnESTR ≤K)	0.121 *	0.146	−0.471 ***
	(1.766)	(1.539)	(−5.920)
lnAging (lnESTR ≥ K)	0.162 **	0.160*	−0.312 ***
	(2.343)	(1.669)	(−4.192)
lnPOP	2.293 ***	−0.912 *	1.149 ***
	(11.366)	(−1.824)	(4.410)
lnrgdp	2.297 ***	−0.912 *	1.164 ***
	(11.353)	(−1.820)	(4.468)
lnEMSTR	−0.299	−0.499 ***	−0.049
	(−1.491)	(−3.598)	(−1.241)
lnINCOM	2.577 ***	5.177 ***	1.004 ***
	(6.776)	(7.896)	(2.696)
ln^2^INCOM	−0.220 ***	−0.189 ***	−0.097 ***
	(−10.120)	(−7.843)	(−5.208)
_cons	−37.641 ***	−6.769	−12.193 ***
	(−11.167)	(−1.359)	(−3.988)
N	220.000	160.000	240.000
R^2^	0.991	0.976	0.987
F	197.614	214.526	260.036

Note: * indicates 10% significance level, ** indicates 5%, *** indicates 1%; _cons represents the constant term of the regression equation; the numeric in brackets indicates the standard deviation of the regression analysis; “lnx” means the logarithm of the variable “x”, “ln^2^x” means the squared term of the variable “lnx”.

**Table 12 ijerph-20-01716-t012:** Threshold effect of technological innovation level on the impact of population aging on carbon emissions in the eastern, central and western regions.

Variables	The East	The Middle	The West
lnAging (lnTECH ≤K)	−0.002	0.084	−0.215 **
	(−0.028)	(0.922)	(−2.504)
lnAging (lnTECH ≥ K)	0.068	0.142	−0.157 *
	(0.989)	(1.576)	(−1.865)
lnPOP	2.143 ***	−1.161 **	1.158 ***
	(10.438)	(−2.412)	(3.790)
lnrgdp	2.136 ***	−1.168 **	1.152 ***
	(10.350)	(−2.424)	(3.770)
lnEMSTR	−0.293	−0.411 ***	−0.085 *
	(−1.491)	(−3.065)	(−1.884)
lnINCOM	2.371 ***	4.881 ***	2.338 ***
	(6.177)	(7.846)	(5.927)
ln^2^INCOM	−0.200 ***	−0.163 ***	−0.160 ***
	(−8.926)	(−6.793)	(−8.007)
_cons	−34.497 ***	−2.028	−19.479 ***
	(−9.831)	(−0.410)	(−5.628)
N	220.000	160.000	240.000
R^2^	0.990	0.982	0.983
F	204.159	235.625	185.026

Note: * indicates 10% significance level, ** indicates 5%, *** indicates 1%; _cons represents the constant term of the regression equation; the numeric in brackets indicates the standard deviation of the regression analysis; “lnx” means the logarithm of the variable “x”, “ln^2^x” means the squared term of the variable “lnx”.

**Table 13 ijerph-20-01716-t013:** The threshold effect of energy consumption structure on the effect of the population aging squared term on carbon emissions.

Variables	National-(1)	National-(2)
ln^2^Aging (lnESTR ≤K)	−0.044 ***	−0.042 ***
	(−4.375)	(−3.272)
ln^2^Aging (lnESTR ≥ K)	0.004	0.013
	(0.452)	(1.171)
lnPOP	1.592 ***	0.805 ***
	(10.492)	(4.618)
lnrgdp	1.602 ***	0.791 ***
	(10.566)	(4.542)
lnEMSTR	−0.081 **	−0.052
	(−1.983)	(−1.388)
lnINCOM	0.731 ***	0.386 *
	(4.134)	(1.908)
ln^2^INCOM	−0.104 ***	−0.065 ***
	(−9.206)	(−4.894)
		(5.554)
Time	NO	YES
_cons	−17.972 ***	−4.717 *
	(−8.773)	(−1.903)
N	620.000	620.000
R^2^	0.988	0.976
F	536.803	195.676

Note: * indicates 10% significance level, ** indicates 5%, *** indicates 1%; _cons represents the constant term of the regression equation; the numeric in brackets indicates the standard deviation of the regression analysis. “National- (1)“ and “National- (2)“ indicate that the empirical analysis sample is the total sample at the national level; “lnx” means the logarithm of the variable “x”, “ln^2^x” means the squared term of the variable “lnx”.

**Table 14 ijerph-20-01716-t014:** Threshold effect of technological innovation level on the impact of the population aging squared term on carbon emissions.

Variables	National-(1)	National-(2)
ln^2^Aging (lnTECH ≤ K)	−0.002	−0.018
	(−0.232)	(−1.637)
ln^2^Aging (lnTECH ≥ K)	0.018 *	0.002
	(1.637)	(7.697)
lnPOP	2.112 ***	0.644 ***
	(13.286)	(3.760)
lnrgdp	2.102 ***	0.635 ***
	(13.192)	(3.704)
lnEMSTR	−0.152 ***	−0.028
	(−3.474)	(−0.759)
lnINCOM	1.673 ***	−0.384 *
	(8.036)	(−1.852)
ln^2^INCOM	−0.173 ***	−0.019
	(−13.370)	(−1.488)
_cons	−29.479 ***	0.957
	(−13.719)	(0.399)
Time	NO	YES
N	620.000	620.000
R2	0.991	0.972
F	446.314	194.549

Note: * indicates 10% significance level, *** indicates 1%; _cons represents the constant term of the regression equation; the numeric in brackets indicates the standard deviation of the regression analysis. “National- (1)“ and “National- (2)“ indicate that the empirical analysis sample is the total sample at the national level; “lnx” means the logarithm of the variable “x”, “ln^2^x” means the squared term of the variable “lnx”.

## Data Availability

The data presented in this study are from the National Bureau of Statistics of China, 2000–2019. (https://data.stats.gov.cn/easyquery.htm (accessed on 9 January 2022)).

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
