# Peer review of "Impact of Population Aging on Carbon Emissions in China: An Empirical Study Based on a Kaya Model"

_ijerph, 2023, doi:10.3390/ijerph20031716_

Round 1

Reviewer 1 Report

The work investigates how China's aging population affects carbon emissions from the production and consumption side using an improved kaya model. The topic is of interest to the scientific community. Here, I provide my major and minor comments, hoping that will help to improve the study.

1. The authors emphasize that the innovation of this paper is to consider the production and consumption sides simultaneously. However, the importance of production and consumption is not well described in Section Introduction. I think that the introduction should be reorganized.

2. The elaboration of scientific problems is the key in the section Introduction. Thus, it is not suitable to describe the kaya model more.

3. The authors gave the improved kaya model, however, where does the kaya model improve?

4. What are the variables on the production and consumption sides that the authors must explicitly point out?

5. In section 2.3.2 Data, the authors should specify the default values in detail, and how to use the K-nearest neighbor filling method to fill the default values.

6. The strict balanced panel data is obtained. I want to know whether spatial relationships among provinces are taken into account.

7. In section 3 Results, many derivative models are given based on the basic model. These derivative models should be placed in the Section method.

8. The authors construct different models to analyze the impact of aging, energy consumption structure, and technological innovation on carbon emissions. Why is not the analysis conducted using a basic model that considers all possible variables?

9. The authors point out that energy consumption structure and technological innovation thresholds exist for the impact of population aging on carbon emissions, with thresholds of 3.275 and 8.904, respectively. please explain what the threshold values mean.

10. In section 4.2.1, this section replaced the independent variable and conducted a control test on the time effect. How to replace the independent variable? And whether the spatial effect should also be considered or analyzed.

11. Although the authors give some description of the results, I believe that some discussion on the results should be necessary.

12. What is the meaning of the National-(1) or National-(2)?

13. line 78, what is the OECD?

14. line 94, the cited reference should be standard.

15. The text in Figure 3 should be larger.

16. line 206, Where, Control in Equation (5) contains …?

17. What is the R&D?

18. What do the three asterisks stand for in Tables 3-13?

19. line 408, the value of 0.0154 should be replaced by 0.154?

20. line 492, the value should be -0.018?

Author Response

Response to Reviewer 1 Comments

Point 1. The authors emphasize that the innovation of this paper is to consider the production and consumption sides simultaneously. However, the importance of production and consumption is not well described in Section Introduction. I think that the introduction should be reorganized.

Response 1:Thank you very much for your suggestion. Based on your suggestion, I have added to the introduction of the article to emphasize the importance of production and consumption itself for economic growth. At the same time, the possible impact of population aging on carbon emissions through the production and consumption side is elaborated. The details are as follows: In the context of China's high-quality economic development, economic growth must rely on productivity improvements, especially total factor productivity, to maintain a reasonable growth rate for China's economy. However, China's ageing population is expected to exceed 300 million in 2025, accounting for more than 20% of the total population and entering a moderate ageing phase, which will have an impact on labor supply and labor productivity, resulting in China's current total factor productivity being only about 40% of that of the US. Low productivity will not only severely constrain China's ability to innovate in energy saving and emission reduction technologies, but will also exacerbate carbon emissions in the production process of enterprises. Actually, the ageing of the population has implications not only for the supply of labor but also for the propensity of people to consume on the demand side. The reason for this is that an ageing society is usually a low-desire society, which can place important constraints on consumption and curb carbon emissions from the consumption side.

Point 2. The elaboration of scientific problems is the key in the section Introduction. Thus, it is not suitable to describe the kaya model more.

Response 2:Thank you very much for your suggestion. Based on your suggestion, I have streamlined the description of the KAYA model by focusing the introductory section of the article more on the statement of the scientific problem under study. Since the introduction section contains a review of existing research, some textual descriptions related to the applied research on the KAYA model are still retained in the latter part of the introduction. Because, I believe that the KAYA model is the core analytical method of this paper, it is necessary to be familiar with the existing application studies of the model.

Point 3. The authors gave the improved kaya model, however, where does the kaya model improve?

Response 3:Thank you very much for your suggestion. An improved KAYA model that compared with traditional kaya model, it has advantages that it can more accurately and comprehensively analyze the influence and marginal effect of different factors on carbon emissions, such as population aging structure, energy consumption structure, employment structure and technological innovation level are considered in the regression model. Which is more in line with the realities of economic and social development. Please see lines 182-222 of the article for specific modification instructions.

Point 4. What are the variables on the production and consumption sides that the authors must explicitly point out?

Response 4:Thank you very much for your suggestion. Regarding your question about the variables of production and consumption, lines 254-262 of the article provide a detailed explanation. The variables of production and consumption are explanatory variables of mechanism analysis, including technological innovation level (TECH) and energy consumption structure (ESTR). Among them,the TECH index is calculated collectively. Firstly, the eight three-level indicators (Table 1) were treated dimensionless. Secondly, the entropy weight method is used to calculate the weight of each three levels. Finally, the weighted average of each index is calculated. The energy consumption structure (ESTR) was characterized by the ratio of the energy consumption of daily necessities to energy consumption of healthcare products. The total energy consumption (ECON) was characterized by the total energy consumption of each province and city (ten thousand tons of standard coal).

Point 5. In section 2.3.2 Data, the authors should specify the default values in detail, and how to use the K-nearest neighbor filling method to fill the default values.

Response 5:Thank you very much for your suggestion. I have elaborated the method of filling in the missing values in the "Data" section of the article according to your suggestion. The details of the improvement are as follows: Owing to some of the missing value types in the paper are non-deterministic values, so the missing values are filled in by calculating similar mean values based on the statistical principle of plurality and the K-nearest neighbor filling method. The specific steps are as follows: First, classify the known values; second, calculate the difference between the maximum and minimum values and divide by 3; third, divide the values into 3 major categories based on the size of the difference between the values; fourth, group the missing values into the category with the largest number of samples based on the principle of plurality; fifth, calculate the mean value of the data belonging to the category, which is the missing value.

Point 6. The strict balanced panel data is obtained. I want to know whether spatial relationships among provinces are taken into account.

Response 6:Thank you very much for your suggestion. Indeed, there are significant differences between different provinces and cities in China in terms of economic level, technological innovation capacity, consumption level and population aging. There must also be large differences between population aging and carbon emissions among different regions. Therefore, regarding spatial relationships among provinces, the heterogeneity of eastern, central, and western regions was analyzed in the course of the latter empirical analysis, based on the level of economic and social development in China.

Point 7. In section 3 Results, many derivative models are given based on the basic model. These derivative models should be placed in the Section method.

Response 7:Thank you very much for your suggestion. According to your advices, I have adjusted the position of the derived model to improve the readability of the article. After adjustment, each subsection of the empirical analysis has a corresponding theoretical model, and details are marked in red in the text.

Point 8. The authors construct different models to analyze the impact of aging, energy consumption structure, and technological innovation on carbon emissions. Why is not the analysis conducted using a basic model that considers all possible variables?

Response 8: Thank you very much for your suggestion. Regarding “the construction of different models to analyze the impact of aging, energy consumption structure, and technological innovation on carbon emissions” , the article is mainly intended to answer the following three questions: first, to verify the average impact of population aging on carbon emissions; second, to test the impact of population aging on carbon emissions through the consumption side; and third, to test the impact of population aging on carbon emissions through the production side. If all variables are included in one analytical model, the impact of population aging on carbon emissions through the consumption side and the production side cannot be identified. It also cannot do further analysis to test the threshold effect on the production and consumption side.

Point 9. The authors point out that energy consumption structure and technological innovation thresholds exist for the impact of population aging on carbon emissions, with thresholds of 3.275 and 8.904, respectively. please explain what the threshold values mean.

Response 9: Thank you very much for your suggestion. The threshold can be seen as dividing the threshold regression model into two parts, and the two parts in which X (the explanatory variable) will have different effects on Y (the explained variable). The threshold model is usually used to analyze the existence of a nonlinear relationship between X and Y in regression models, of which the Kuznets curve is the most typical. In the text, I provide a brief explanation of the meaning of thresholds in the threshold model analysis section, which is marked in red in the text. It can be further explained as follows: In this research, the consumption structure value is 3.275, which indicates that the aging of the population has a different degree of impact on carbon emissions on both sides of the consumption structure around the threshold (3.275). Similarly, when the value of the comprehensive technological innovation index reaching 8.904, the impact coefficient of population aging on carbon emissions will change, which indicates that the aging of the population has a different degree of impact on carbon emissions on both sides of technological innovation around the threshold.

Point 10. In section 4.2.1, this section replaced the independent variable and conducted a control test on the time effect. How to replace the independent variable? And whether the spatial effect should also be considered or analyzed.

Response 10: Thank you very much for your suggestion. Substitution of appropriate independent variables is one of the effective analytical methods for regression model robustness testing, which can test the variable selection bias problem. Regarding "how to replace the independent variables", this study uses the quadratic term of population aging as a replacement variable for population aging, because both population aging and the squared term of population aging can respond to the level of population aging to some extent. Regarding “whether spatial effects are considered”, the article does a subsample and spatial heterogeneity test analysis for the East, Central and West regions in the previous subsection content.

Point 11. Although the authors give some description of the results, I believe that some discussion on the results should be necessary.

Response 11: Thank you very much for your suggestion. Based on your suggestion that "some discussion on the results should be necessary", I further elaborated on the possible causes of the empirical test results. The specific additions are highlighted in red in the text. For example, lines 426-429, 436-441, and 473-479.

Point 12. What is the meaning of the National-(1) or National-(2)?

Response 12: Thank you very much for your suggestion. National (1) and National (2) indicate that the empirical analysis sample is the total sample at the national level, which is distinguished from the analysis of regional heterogeneity (East, Central, and West regions) in the later section. where National (1) does not control for time variables and National (2) controls for time variables.

Point 13. line 78, what is the OECD?

Response 13: Thank you very much for your suggestion. OECD is an abbreviation for organization for economic cooperation and development. The corresponding content in the article is intended to express the OECD countries. To make the article more readable, I have added the full name for clarification. The specific modifications are as follows:Jiang [9] used kaya model empirically tested the relationship between population structure and carbon emissions with life-cycle theory based on cross-country panel data of 26 OECD (organization for economic cooperation and development) member countries.

Point 14. line 94, the cited reference should be standard.

Response 14: Thank you very much for your suggestion. A standard citation of the literature was made based on your prompt. The specific modifications are as follows: Shen [15] found the same conclusion through a systematic generalized moments estimation econometric model analysis, based on interprovincial panel data for China from 1995 to 2012.

Point 15. The text in Figure 3 should be larger.

Response 15: Thank you very much for your suggestion. I have enlarged the text content of Figure 3 appropriately and adjusted the pitures combination form to reflect the meaning of the image content more clearly. The specific adjustments are as follow pictures:

Point 16. line 206, Where, Control in Equation (5) contains …?

Response 16: Thank you very much for your suggestion. Control in Equation (5) contains income level (INCOM) and its square term, employment structure (EMSTR), gross product per capita (rgdp) and population size (POP), which I explained it in the variables section of the original version. However, in order to make the meaning of "Control" clearer to the reader, I have added a brief explanation below equation (5) (line 219 in new version). Revised as follow screenshot:

Point 17. What is the R&D?

Response 17: Thank you very much for your suggestion. R&D is the abbreviation for research and development in China, which is mainly used in this study to denote the investment in scientific research. In order to make the meaning of "R&D" clear to the reader, I have explained it in the text.

Point 18. What do the three asterisks stand for in Tables 3-13?

Response 18: Thank you very much for your suggestion. I have explained the meaning of “***” in corresponding tables. Specifically, “*”indicates 10% significance level, “**” indicates 5%, “***” indicates 1%.

Point 19. line 408, the value of 0.0154 should be replaced by 0.154?

Response 19: Thank you very much for your suggestion. Yes, the value in line 408 (in original version) should be -0.154 and has been corrected in the text. Revised as follow screenshot:

Point 20. line 492, the value should be -0.018?

Response 20: Thank you very much for your suggestion. Yes, the value in line 492 (in original version) should be -0.018 and has been corrected in the corresponding place. Revised as follow screenshot:

Reviewer 2 Report

“Impact of population aging on carbon emission in China: An empirical study based on kaya model” aims to explore the impact of population aging on carbon emissions in China from the perspective of the production and consumption side. The paper has a high quality in publication, but some information should give more information.

Introduction section

- The author should give a bit of talking about why aging-related to carbon emissions apart from the increasing number of aging in China, or the evidence to support the consumption of aging compared to other ages.

Materials and Methods section

- In figure 2, the author should use the same color each year for all regions.

- In figure 3, the small pictures in the bottom right are not clear. Can they be deleted or in a bigger size?

- In figure 3, the author should give identification of the white color in Tibet.

- Link 156 “Figure 2-2000 transforms into Figure 3-2019”, please check whether it is mistyping or not.

Result section

- In the Descriptive statistical analysis part, the correlation analysis should be used to confirm the relationship between carbon dioxide emission and other variables.

- In tables 3 -5 and 8, the author should give more clarification on the meaning of the numeric indicated in the table. What is the meaning of “***”, “_cons” and the numeric in brackets?

Both the result and discussion sections

- The author should add references to support your result or your observation such as “Observing the significance level of regression parameters, it can be found that the effect between energy consumption structure and carbon emissions is not significant, and the reason is that carbon emissions will be more affected by the consumption behavior of economic subjects.”. The author should find the reference to support the behavior of economic subjects having more effects on carbon emissions than energy consumption did.

Author Response

Response to Reviewer 2 Comments

Introduction section

Point 1: The author should give a bit of talking about why aging-related to carbon emissions apart from the increasing number of aging in China, or the evidence to support the consumption of aging compared to other ages.

Response 1: Thank you very much for your suggestion, the following has been added to the introduction section of the article: In the context of China's high-quality economic development, economic growth must rely on productivity improvements, especially total factor productivity, to maintain a reasonable growth rate for China's economy. However, China's ageing population is expected to exceed 300 million in 2025, accounting for more than 20% of the total population and entering a moderate ageing phase, which will have an impact on labor supply and labor productivity, resulting in China's current total factor productivity being only about 40% of that of the US. Low productivity will not only severely constrain China's ability to innovate in energy saving and emission reduction technologies, but will also exacerbate carbon emissions in the production process of enterprises. Actually, the ageing of the population has implications not only for the supply of labor but also for the propensity of people to consume on the demand side. The reason for this is that an ageing society is usually a low-desire society, which can place important constraints on consumption and curb carbon emissions from the consumption side.

Materials and Methods section

- Point2: In figure 2, the author should use the same color each year for all regions.

Response 2: Thank you very much for your suggestion, the changes have been made according to your request and the results are as follows. The colors have been standardized for the same year in all regions.

Figure 2. Ageing Dependency Ratio and Carbon Emissions in East Midwest Region, 2000/2010/2019.

- Point 3: In figure 3, the small pictures in the bottom right are not clear. Can they be deleted or in a bigger size?

Response 3: Thank you very much for your suggestion. The small pictures in the bottom right corner of Figure 3 does not have a substantial economic meaning in this study; it serves to avoid the political errors associated with the map boundaries. Therefore, enlarging the small pictures in the bottom right corner may reduce the representation of the main message of the study analysis; if it is removed, it may introduce political errors.

- Point 4: In figure 3, the author should give identification of the white color in Tibet.

Response 4: Thank you very much for your suggestion. Because the economic and social reality of Tibet differs considerably from other provincial areas of China and relevant statistics are missing, Tibet was not included as a subject of study in the thesis. However,Tibet as a part of China, must be included in the mapping. If Tibet were to be filled in white, the complexity of the information in the color indicators in the bottom left corner would increase, which makes the article more difficult to understand without having practical implications, so Tibet was not color filled. In fact, the existing Figure 3 Tibetan area has a white base color and its representation is consistent with your suggestion.

-Point 5: Link 156 “Figure 2-2000 transforms into Figure 3-2019”, please check whether it is mistyping or not.

Response 5: Thank you very much for correcting the detail error in the article, the correct expression should be “Figure 3-2000 transforms into Figure 3-2019” and has been corrected at the corresponding place in the article.

Result section

-Point 6: In the Descriptive statistical analysis part, the correlation analysis should be used to confirm the relationship between carbon dioxide emission and other variables.

Response 6: Thank you very much for your suggestion, which tightens the framework of the logical analysis of the article, for which I added correlation analysis between variables in the descriptive statistics section. The specific additions are as follows: To ensure the reliability of the results of the subsequent empirical analysis, this section further uses stata17.0 software to test the correlations between carbon emissions and population aging and other variables. The test results show that carbon emissions are significantly correlated with the variables of population aging, per capi-ta economic development level, population size, total energy consumption, and energy consumption structure with a significance level of 1%.

Table 3. Correlation analysis of variables

Variables

(1)

(2)

(3)

(4)

(5)

(6)

(7)

(8)

(9)

(10)

(11)

(1) lnCO2

1.000

(2) lnAGING

0.306

1.000

(0.000)

(3) ln2AGING

0.317

0.997

1.000

(0.000)

(0.000)

(4) lnrgdp

0.163

0.348

0.351

1.000

(0.000)

(0.000)

(0.000)

(5) lnPOP

0.282

0.279

0.270

-0.482

1.000

(0.000)

(0.000)

(0.000)

(0.000)

(6) lnTECH

0.399

0.591

0.589

0.590

0.366

1.000

(0.000)

(0.000)

(0.000)

(0.000)

(0.000)

(7) lnESTR

0.025

0.374

0.356

0.095

0.384

0.428

1.000

(0.539)

(0.000)

(0.000)

(0.018)

(0.000)

(0.000)

(8) lnECON

0.152

0.429

0.414

0.229

0.464

0.631

0.832

1.000

(0.000)

(0.000)

(0.000)

(0.000)

(0.000)

(0.000)

(0.000)

(9) lnEMSTR

-0.040

0.336

0.343

0.511

-0.091

0.506

0.217

0.211

1.000

(0.320)

(0.000)

(0.000)

(0.000)

(0.023)

(0.000)

(0.000)

(0.000)

(10) lnINCOM

0.263

0.460

0.468

0.714

0.011

0.792

0.210

0.318

0.736

1.000

(0.000)

(0.000)

(0.000)

(0.000)

(0.785)

(0.000)

(0.000)

(0.000)

(0.000)

(11) ln2INCOM

0.256

0.461

0.470

0.715

0.010

0.791

0.210

0.315

0.741

0.999

1.000

(0.000)

(0.000)

(0.000)

(0.000)

(0.808)

(0.000)

(0.000)

(0.000)

(0.000)

(0.000)

Note: The content in parentheses indicates the significance level.

- Point 7: In tables 3 -5 and 8, the author should give more clarification on the meaning of the numeric indicated in the table. What is the meaning of “***”, “_cons” and the numeric in brackets?

Response 7: Thank you very much for your suggestion, I have explained the meaning of “***” “_cons” and the contents of the brackets at the bottom of the table3-5 and 8. “*”indicates 10% significance level, “**” indicates 5%, “***” indicates 1%; “_cons” represents the constant term of the regression equation; the numeric in brackets indicates the standard deviation of the regression analysis.

Both the result and discussion sections

- Point 8: The author should add references to support your result or your observation such as “Observing the significance level of regression parameters, it can be found that the effect between energy consumption structure and carbon emissions is not significant, and the reason is that carbon emissions will be more affected by the consumption behavior of economic subjects.”. The author should find the reference to support the behavior of economic subjects having more effects on carbon emissions than energy consumption did.

Response 8: Thank you very much for your suggestion, which will enhance the normality of the article. For this reason, I have added references in this section that can support the findings of this paper, and the added references are marked in red in the body of the paper.

Round 2

Reviewer 1 Report

The authors have successfully addressed the comments and the paper may be accepted.